# Investigating the Robustness of a Rodent “Double Hit” (Post-Weaning Social Isolation and NMDA Receptor Antagonist) Model as an Animal Model for Schizophrenia: A Systematic Review

**DOI:** 10.3390/brainsci13060848

**Published:** 2023-05-24

**Authors:** Khanyiso Bright Shangase, Mluleki Luvuno, Musa V. Mabandla

**Affiliations:** Department of Human Physiology, School of Laboratory Medicine and Medical Science, College of Health Sciences, University of KwaZulu-Natal, Durban 4041, South Africa; luvunom@ukzn.ac.za (M.L.); mabandlam@ukzn.ac.za (M.V.M.)

**Keywords:** post-weaning social isolation (SI), NMDA receptor antagonist, ketamine, phencyclidine (PCP), dizocilpine (MK801), animal models, basic research, rodents, schizophrenia

## Abstract

Schizophrenia is a debilitating psychiatric disorder comprising positive, negative, and cognitive impairments. Most of the animal models developed to understand the neurobiology and mechanism of schizophrenia do not produce all the symptoms of the disease. Therefore, researchers need to develop new animal models with greater translational reliability, and the ability to produce most if not all symptoms of schizophrenia. This review aimed to evaluate the effectiveness of the rodent “double hit” (post-weaning social isolation and N-methyl-D-aspartate (NMDA) receptor antagonist) model to produce symptoms of schizophrenia. This systematic review was developed according to the 2020 PRISMA guidelines and checklist. The MEDLINE (PubMed) and Ebscohost databases were used to search for studies. The systematic review is based on quantitative animal studies. Studies in languages other than English that could be translated sufficiently using Google translate were also included. Data extraction was performed individually by two independent reviewers and discrepancies between them were resolved by a third reviewer. SYRCLE’s risk-of-bias tool was used to test the quality and biases of included studies. Our primary search yielded a total of 47 articles, through different study selection processes. Seventeen articles met the inclusion criteria for this systematic review. Ten of the seventeen studies found that the “double hit” model was more effective in developing various symptoms of schizophrenia. Most studies showed that the “double hit” model is robust and capable of inducing cognitive impairments and positive symptoms of schizophrenia.

## 1. Introduction

Schizophrenia is a severe psychiatric disorder that is generally characterized by profound impairment in thinking which affects language, sense of self, and perception [1]. Schizophrenia affects more than 21 million individuals globally [1]. The symptoms of the disease are divided into three categories, viz., positive, negative, and cognitive [2,3]. Positive symptoms include hallucinations and delusions, and negative symptoms include avolition, deficits in social functioning, anhedonia, and blunted affects, while cognitive symptoms are deficits in attention and perception, in executive control, and in working and long-term memory [4,5]. Schizophrenia onset usually manifests during post adolescence (16–25 years) [6]. Males usually show significantly higher incidences of schizophrenia than females [7]. Schizophrenia treatment focuses on the stage when the affected patients show clear clinical symptoms, such as psychosis [8]. The well-documented mechanisms of action for the antipsychotic treatment involve correcting the high levels of dopamine turnover in the striatum [9]. Schizophrenia patients at the first episode of the disorder respond reasonably well to treatment; the challenge is to keep them in a good state [10,11]. Previous studies have proposed that D-serine and glycine can improve negative and positive symptoms, but this has not been observed for cognitive impairments [12,13]. α 2/3-selective agonist and α 5-selective inverse agonist are selective GABAergic drugs that have been proposed to improve cognition in schizophrenia patients [8]. Animal models of schizophrenia are used to study multifactorial psychiatric disorders including the neurobiological basis of different disorders [6].

When compared to humans, animal models produce a speedy platform to examine structural and molecular alterations that trigger the cause of the pathology, and to test innovative therapeutics that are not possible to investigate in humans [6]. There are more than 20 different animal models of schizophrenia, and they are divided into four categories: developmental, lesion, genetic manipulation, and drug induced [6]. Rodents are social mammals and housing in a group or in isolation has been shown to affect behaviour [14,15]. Post-weaning social isolation of rats by placing them individually results in social deprivation that causes permanent changes in brain development, which may lead to behavioural deficits in adulthood [16,17]. Post-weaning social isolation in rodents produces spontaneous hyper-locomotor activity, sensorimotor gating deficits, enhanced response to novelty, heightened anxiety states, aggression, and cognitive impairment [17,18,19,20,21,22]. All these changes are collectively called isolation syndrome, and some resemble the positive symptoms of schizophrenia. When placed in an aversive novel arena, socially isolated rats are more active than their non-isolated counterparts [23,24]. The robustness of social isolation can be reduced by changing the type of rat strain, gender, caging condition, and routine handling [25]. Positive symptoms can be treated with antipsychotic drugs, such as haloperidol and olanzapine, while negative and cognitive symptoms remain resistant to the available antipsychotic treatment [6].

Studies have shown that glutamatergic pathway dysfunction is a fundamental pathological shift seen in schizophrenia [26,27,28]. The N-methyl-D-aspartate receptor (NMDAR) has been shown to play a role in the pathophysiology of schizophrenia [28]. Treating rats with NMDA glutamate receptor antagonists (PCP or MK801 or ketamine) induces several behavioural abnormalities such as impairments in reversal learning, social interaction, prepulse inhibition (PPI), and working memory [29,30,31,32]. These are similar to those observed in schizophrenia. Most rodent models of schizophrenia tend to replicate aspects of the positive symptoms. Therefore, there is an urgent need for the development of a schizophrenic animal model that will adequately replicate positive, negative, and cognitive symptoms of the disorder. Researchers have started designing the “double hit” model by combining post-weaning social isolation and an NMDA receptor antagonist to develop a strong animal model of schizophrenia that has the potential to replicate positive, negative, and cognitive symptoms [33,34,35].

### 1.1. Study Aim

The present review aimed to evaluate work published on the effectiveness of the “double hit” model in producing all the symptoms of schizophrenia.

### 1.2. Study Objectives

To determine the effectiveness of “double hit” model of schizophrenia in producing positive, negative, and cognitive symptoms of schizophrenia.

To assess the strength of the “double hit” model of schizophrenia as a reliable developmental model of schizophrenia.

## 2. Methods

This systematic review is registered at the International Prospective Register of Systematic reviews (PROSPERO) (CRD42021247585). This systematic review was developed according to the preferred reporting items for systematic review and meta-analysis (PRISMA) guidelines and checklist of 2020 [36].

### 2.1. Search Strategy

Medical search headings (MeSH) were used in the formulation of a systematic search strategy. PubMed and Ebscohost databases were utilized to search for published studies. The literature search focused on the English language only and animals (rodent) subjects; all databases were searched until 30 May 2023. All included study reference lists were scanned to confirm the literature saturation. Lastly, the bibliography of all the included studies was circulated to the systematic review team and other schizophrenia experts chosen by the team. The literature search was undertaken by two independent authors (KBS and ML) and another (MM) was approached for arbitration. The literature search was based on these keywords and rodent subject headings: “schizophrenia”, “social isolation”, “NMDA receptor antagonists”, “basic research”, and “animal models”.

### 2.2. Selection Process

The systematic review is based on quantitative animal studies. Studies in other languages that were successfully and sufficiently translated to English by Google Translate were also included. The process of screening studies was conducted by two authors independently (KBS and ML) to eliminate any inconsistencies in terms of the fitness of studies. The screening of studies was performed in the following order: title, abstract, keywords and synonyms followed by full-text screening. All studies that induce schizophrenic-like symptoms using a “double hit” rodent model (post-weaning social isolation and NMDA receptor antagonist) were included. Any disagreements between two authors (KBS and ML) were resolved by allowing the third author (MM) to screen those studies and then after discussion an agreement was reached. The PRISMA flow diagram was utilized to document the final selection process. All included studies were subjected to data collection, and quality assessment.

### 2.3. Data Extraction

Data extraction was executed by two authors independently (KBS and ML), and the third author (MM) was approached in case of any disagreements between two authors. All important information, such as the authors, country, year of publication, study design, sample size, rodent characteristics (strain, gender, and age), NMDA receptor antagonists used and dosage, period of isolation, symptom severity, types of control used, and control of symptoms if included, were recorded in a Microsoft Excel table created by two authors (KBS and ML). The third author (MM) read and approved the content of this Microsoft Excel table. Included studies consisted of different social isolation periods, different NMDA receptor antagonists, and different NMDA receptor antagonist dosages. Groups from various arms of the study were merged into a single group to avoid the possibility of introducing bias caused by multiple statistical comparisons with one control group.

### 2.4. Risk of Bias and Quality Assessment

SYRCLE’s risk-of-bias tool was utilized to test the quality and possibility of bias in all included studies [37]. The SYRCLE risk-of-bias tool is a derived version of the Cochrane risk-of-bias tool. Evidence from the included studies was critically appraised by the SYRCLE risk-of-bias tool. SYRCLE risk of bias is made up of ten entries, and these entries answer questions that are related to the following topics: selection bias, detection bias, performance bias, reporting bias, attrition bias, and other biases [37]. To assist in reaching a strong judgment, additional signalling questions are included. “Yes” means a low risk of bias, “no” means high risk of bias, and “unclear” means an unclear risk of bias [37]. If one of the questions is answered with “no,” this means a high risk of bias for that entry [37]. Two independent authors (KBS and ML) evaluated the quality of each included study. In case of disagreements between two authors, the third author (MM) was asked to adjudicate.

## 3. Results

### 3.1. Study Selection

Our primary search yielded a total of 47 articles (Figure 1). Through different study selection processes, 17 articles met the inclusion criteria for this systematic review [33,34,35,38,39,40,41,42,43,44,45,46,47,48,49,50,51]. Eight studies used Sprague Dawley rats [34,39,40,41,42,43,44,48], three studies used Lister-hooded rats [46,47,49], three studies used Wistar rats [38,45,50], and another three studies used mice [33,35,51]. All studies used male animals except one that used both male and female animals [45]. Most studies originated from Europe; the United Kingdom [46,47,49], and Spain [33,35,51] each had three studies, Hungary [38,45] had two, while France [50] and Germany [44] had one each. North America (Canada) produced five papers [39,40,41,42,43], and Asia (China) two papers [34,48].

### 3.2. Risk of Bias and Quality Assessment

The quality scores of each study assessing the risk of bias are displayed in Table 1. All studies showed a high level of quality. There is one study with “no” response to the question; this response negatively affected the quality of the study [48]. All studies included the control group or animals without treatment (group housed + saline/vehicle), and all studies used the control group as the normal group.

### 3.3. Effectiveness and Robustness of the “Double Hit” (Post-Weaning Social Isolation and NMDA Receptor Antagonist) Model of Schizophrenia

Ten studies found that the “double hit” model is more effective when compared to the intervention using social isolation on its own or an NMDA receptor antagonist [33,35,40,44,45,46,47,49,50,51]. Three of these studies were able to produce positive symptoms of schizophrenia, which is the impairment on the locomotor activity [40,47,50]. Other studies showed that the “double hit” model produced impairments on the novel object discrimination test [46,47]; furthermore, another study showed impairment on the social recognition test [50], and these are related to cognitive symptoms of schizophrenia. Another study reported the presence of heat shock protein 70 in brain regions of animals treated with the “double hit” model of schizophrenia [44]. The “double hit” model induces reductions in prepulse inhibition and reduced cingulate 1 cortex volume [35]. Seven studies did not find the “double hit” model to be more effective when compared to each treatment alone [34,38,39,41,42,43,48].

### 3.4. The Effect of “Double Hit” (Post-Weaning Social Isolation and NMDA Receptor Antagonist) Model of Schizophrenia on Neurotransmitters

One study reported disturbance in excitatory/inhibitory neurotransmitter balance in the key brain region [33]. The study of Shortall et al. [49] reported reduced glutamate release. Two studies reported decreased hippocampus and prefrontal cortex GABA release [40], and reduced Gad67 expression [35]. Another study reported a significant decrease in the number of PV+ interneurons and perineuronal nets when compared to normal control [51].

### 3.5. Use of “Double Hit” (Post-Weaning Social Isolation and NMDA Receptor Antagonist) Model as a Developmental Model of Schizophrenia

Ten studies found that the “double hit” model is more effective when compared to an intervention using social isolation on its own or an NMDA receptor antagonist [33,35,40,44,45,46,47,49,50,51]. Four of these studies injected phencyclidine (PCP) on PND7, PND9, and PND11 [46,47,49,50], three studies injected MK801 on PND7 [33,35,51]. Post-weaning social isolation period varied per study, but seven studies started social isolation on PND21 [33,35,40,44,49,50,51], while other three studies started the social isolation on PND23 [45,46,47].

**Figure 1 brainsci-13-00848-f001:**
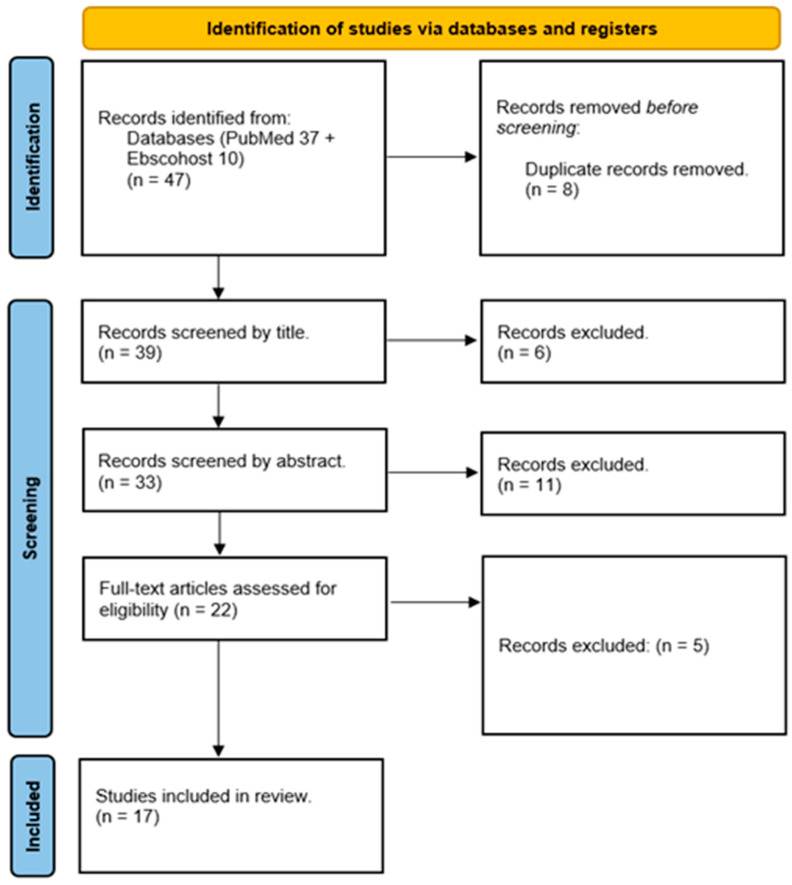
Flowchart of the study selection process and the characteristics of the included studies are summarized in Table 2.

## 4. Discussion

Studies have shown that schizophrenia incidence is significantly higher in males than in females [52,53]. Furthermore, these studies proposed that the overall male:female risk ratio is 1.4:1. This difference could not be explained by methodological factors connected to age variety or diagnostic criteria [7,52,53]. In this review, we showed that most studies used male rodents. We found that from the studies that used rats, 50% of them used Sprague Dawley rats, while 25% of them used Lister hooded rats, and another 25% used Wistar rats. Rats are commonly used as animal models of schizophrenia because they are social animals and their behaviour better mimics the behaviour seen in humans [54]. The main reason why fewer studies have used mice as animal models of schizophrenia could be that sometimes it is difficult to work with mice as they are more aggressive when compared to rats [54]. Different studies used different NMDA receptor antagonists; ten animals used MK801, while four used phencyclidine and two used ketamine. Scientists prefer ketamine over phencyclidine because phencyclidine produces high levels of neurotoxicity and severe hallucinations [55]. Neuroimaging and neuropsychological studies concluded that for ethical reasons MK801 and ketamine are to be preferred over phencyclidine [56]. MK801 produces much longer-lasting symptoms than ketamine and phencyclidine, and the fact that it is safer than phencyclidine is the reason most studies employ it.

### 4.1. “Double Hit” (Post-Weaning Social Isolation and NMDA Receptor Antagonist) Model on Neurotransmitters

A study found an imbalance on excitatory and inhibitory neurotransmission in the prefrontal cortex and amygdala of “double hit” (Mk801-SI) mice [33]. Furthermore, imbalances in excitatory and inhibitory neurotransmission on vital brain regions has been found to be an underlying factor in psychiatric disorders such as schizophrenia [33]. The “double hit” model has also been reported to have an influence in the release of different neurotransmitters. One study on the “double hit” model found a reduction in the number of PV+ interneurons; these are fast-spiking GABAergic neurons that modulate inhibitory control in both cortical and subcortical circuits [51]. They also reported reduced perineuronal nets, which provide a protective layer, maintain optimum local ionic homeostasis, and provide neuronal protection against oxidative stress [51]. A study on the (PCP-SI) model reported the downregulation of several GABA-associated genes, such as the PVALB gene encoding parvalbumin and three GABA receptor subunit genes (GABBR1, GABRA4, GABRB2) [47]. One study proposed that the increase in locomotor activity may be caused by decreased on the inhibitory neurotransmitter (GABA) function in the mesolimbic region and prefrontal cortex [40]. Six genes encoding enzymes that control glutamate metabolism were significantly downregulated in PCP-SI rats [47]. It was not immediately evident if the downregulation of these genes is a compensatory outcome of lower basal glutamate in (PCP-SI) rats or a direct cause of the developmental manipulation [47].

### 4.2. “Double Hit” (Post-Weaning Social Isolation and NMDA Receptor Antagonist) Model on Positive Symptoms of Schizophrenia

Ten studies showed that the “double hit” model produced greater behavioural and physiological deficits than observed with either treatment alone [33,35,40,44,45,46,47,48,49,50]. Three studies found that the “double hit” model caused locomotor hyperactivity, which is a positive symptom of schizophrenia [40,47,50]. Locomotor hyperactivity is expressed as increased horizontal activity and is easily noticeably after the first 15 min in an open field test. This is caused by increase in mesolimbic dopamine and its serves as a baseline for the positive symptoms of schizophrenia [24]. Two studies found that the “double hit” model caused impairment in the PPI, which is linked to sensorimotor gating deficits [35,48]. It has been shown that this PPI impairment is linked to the mesolimbic dopamine system because the injection of 6-hydroxydopamine depletes dopamine in the nucleus accumbens and reverses PPI impairment [57].

### 4.3. “Double Hit” (Post-Weaning Social Isolation and NMDA Receptor Antagonist) Model on Cognitive Symptoms of Schizophrenia

Three studies showed that the “double hit” model (PCP-SI) causes cognitive impairments similar to those observed in patients with schizophrenia [46,49,50]. Two studies found that animals exposed to the “double hit” model failed to identify the difference between a familiar and a novel object in a novel discrimination paradigm [46,49]. The novel object recognition test measures visual recognition memory performance. This has translational relevance to one of the cognitive domains present in schizophrenia [58]. A study reported that “double hit” model (PCP-SI) rats were unable to recognize a juvenile rat contacted 30 min previously, possibly suggesting a lack of motivation to interact socially [50]. It has been reported that in the (MK801-SI) model, rats showed an increased expression of heat shock protein 70, a marker for neuronal damage in the neocortical regions [44]. NMDA receptor antagonist, which is used to induce schizophrenia, has been also proven to induce cortical injury (cell necrosis) in the neocortical regions [59]. Other studies have gone further to investigate if available drugs used to treat schizophrenia will be able to reverse different symptoms of schizophrenia induced by a “double hit” model.

### 4.4. Response of “Double Hit” (Post-Weaning Social Isolation and NMDA Receptor Antagonist) Model on Drugs Used to Reversed Symptoms of Schizophrenia

A study showed that a drug (lamotrigine) was able to reverse hyperactivity caused by PCP-SI on rats placed in a novel arena [47]. Lamotrigine achieves this by reducing excitability in the striatal neurons which cause the inhibition of pre-synaptic voltage-gated sodium channels which reduce glutamate release [60,61]. Lamotrigine at a dose of (10 mg/kg i.p) was able to reverse the PCP-SI induced impairments in a novel object recognition test [47]. Furthermore, another study showed that clozapine was able to reverse locomotor hyperactivity and social recognition impairment caused by PCP-SI [50].

## 5. Conclusions, Limitations, and Future Directions

In conclusion, most studies showed that the “double hit” (post-weaning social isolation and NMDA receptor antagonist) model is robust and capable of inducing a wider spectrum of more severe cognitive impairments and positive symptoms of schizophrenia. The “double hit” model has also proven to be a robust and reliable developmental model of schizophrenia. Different drugs (lamotrigine and clozapine) were able to reverse different impairments that were caused by the “double hit” model; this showed the predictive validity of this model and its potential as a translational model. More research is needed to investigate the effect of the “double hit” model on negative symptoms of schizophrenia. Finally, more research needed to investigate the effect of this “double hit” model on epigenetic markers. According to our knowledge, this is the first study to review the “double hit” models of schizophrenia; this review will contribute positively to the development of effective animal models of schizophrenia that will be able to produce all symptoms of schizophrenia. Nevertheless, there are limitations to our study. We only considered studies on the “double hit” in rodent models of schizophrenia; we excluded other animal and human subjects. There is a possibility that we might have missed some papers because of how specific our search strategy was. The study only focused on the post-weaning social isolation and NMDA receptor antagonist “double hit” model of schizophrenia; there are many possible “double hit” models of schizophrenia that could provide valuable data. Choosing the NMDA receptor antagonist in our “double hit” model meant our paper was more focused on the glutamate neurotransmitter; there are other neurotransmitters that are involved in schizophrenia. Given the limitations of the current study, future study is needed to broaden the spectrum and cover both animal models and human subjects of the available “double hit” studies of schizophrenia. Another study is needed that will cover all “double hit” models, not only postweaning social isolation and the NMDA receptor antagonist.

## Figures and Tables

**Table 1 brainsci-13-00848-t001:** Shows the risk of bias for all selected articles.

Studies	Selection Bias			Performance Bias		Detection Bias		Attrition Bias	Reporting Bias	Other	Judgement/Risk of Bias
	Was the Allocation Sequence Adequately Generated and Applied?	Were the Groups Similar at Baseline or Were They Adjusted for Confounders in the Analysis?	Was the Allocation Adequately Concealed?	Were the Animals Randomly Housed during the Experiment?	Were the Caregivers and/or Investigators Blinded from Knowledge of Which Intervention Each Animal Received during the Experiment?	Were Animals Selected at Random for Outcome Assessment?	Was the Outcome Assessor Blinded?	Were Incomplete Outcome Data Adequately Addressed?	Are Reports of the Study Free of Selective Outcome Reporting?	Was the Study Apparently Free of Other Problems That Could Result in High Risk of Bias?	
[38]	Yes	Yes	Yes	Yes	Yes	Yes	Yes	Yes	Yes	Yes	Low
[39]	Yes	Yes	Yes	Yes	Yes	Yes	Yes	Yes	Yes	Yes	Low
[40]	Yes	Yes	Yes	Yes	Yes	Yes	Yes	Yes	Yes	Yes	Low
[41]	Yes	Yes	Yes	Yes	Yes	Yes	Yes	Yes	Yes	Yes	Low
[42]	Yes	Yes	Yes	Yes	Yes	Yes	Yes	Yes	Yes	Yes	Low
[43]	Yes	Yes	Yes	Yes	Yes	Yes	Yes	Yes	Yes	Yes	Low
[44]	Yes	Yes	Yes	Yes	Yes	Yes	Yes	Yes	Yes	Yes	Low
[45]	Yes	Yes	Yes	Yes	Yes	Yes	Yes	Yes	Yes	Yes	Low
[46]	Yes	Yes	Yes	Yes	Yes	Yes	Yes	Yes	Yes	Yes	Low
[47]	Yes	Yes	Yes	Yes	Yes	Yes	Yes	Yes	Yes	Yes	Low
[48]	Yes	Yes	Yes	Yes	Yes	Yes	Yes	No	Yes	Yes	Low
[33]	Yes	Yes	Yes	Yes	Yes	Yes	Yes	Yes	Yes	Yes	Low
[34]	Yes	Yes	Yes	Yes	Yes	Yes	Yes	Yes	Yes	Yes	Low
[35]	Yes	Yes	Yes	Yes	Yes	Yes	Yes	Yes	Yes	Yes	Low
[49]	Yes	Yes	Yes	Yes	Yes	Yes	Yes	Yes	Yes	Yes	Low
[50]	Yes	Yes	Yes	Yes	Yes	Yes	Yes	Yes	Yes	Yes	Low
[51]	Yes	Yes	Yes	Yes	Yes	Yes	Yes	Yes	Yes	Yes	Low

“Yes” means low risk of bias, “No” means high risk of bias, “Low” is a judgement meaning low risk of bias for the paper.

**Table 2 brainsci-13-00848-t002:** Characteristics for all included studies.

Study	Location of the Study	Year of Publication	Species	Sex	NMDA Receptor Antagonist, Dosage, Period of Injection, Type of Injection	Commencement of Social Isolation, Period of Social Isolation	Research Groups per Study	Outcome of the “Double Hit” Model
[41]	Canada	2012	Sprague Dawley rat	Male	MK801, 0.5 mg/kg, (PND 56–62) 2 times daily for 7 days, i.p. injection	From PND21-PND78	GH + Sal, GH + MK, SI + Sal, and SI + MK.	Combined post-weaning social isolation and subchronic MK801 treatment did not produce additive or synergistic effects on locomotor behaviour or Gaba signalling, but rather induce differential effects on GABAa receptor binding.
[44]	Germany	2013	Sprague Dawley rat	Male	MK801, (2 mg/kg) one injection, on PND64, i.p. injection	On PD21 (weaning age corresponding to pre-adolescence), rats were housed either individually or in groups of three in cages for a 6-week period from PND21 to PND63	GH + Sal, GH + MK, SI + Sal, and SI + MK	Juvenile rats exposed to chronic isolation had increased MK801-triggered expression of heat shock protein 70, a marker of neuronal injury, in the retrosplenial cortex. This suggests an additive effect of juvenile stress and NMDA receptor blockade, with possible relevance for schizophrenia.
[42]	Canada	2012	Sprague Dawley rat	Male	MK801, 0.5 mg/kg, (PND56-PND62) was injected twice daily, i.p. injection.	On PND21-PND73 animals were socially isolated and rats remained in their assigned housing for the duration of the experiment.	GH + Sal + Sal, GH + MK + Sal, SI + Sal + Sal, SI + MK + Sal, GH + Sal + PLZ, GH + MK + PLZ, SI + Sal + PLZ, and SI + MK + PLZ	The combination of social isolation and subchronic MK801 did not produce greater behavioural changes than either treatment alone.
[47]	United Kingdom	2016	Lister-hooded rat	Male	PCP, 10 mg/kg on post-natal days (PND7, PND9, and PND11), s.c. injection	Started on PND 23 till the end of the study	V + GH + V, V + GH + L15, PCP + SI + V, PCP + SI + L10, and PCP + SI + L15	Acute lamotrigine (10–15 mg/kg i.p.) reversed the hyperactivity and novel object recognition impairment induced by “double hit” model (PCP-SI) but had no effect on the prepulse inhibition deficit.
[35]	Spain	2020	GIN mice	Male	MK801, 1 mg/kg on PND 7 pups received one injection, i.p. injection	PND21 till the end of the study-PND133	CTRL + V, CTRL + THC, (SI + MK) + V, and (SI + MK) + THC	We found that “double hit” had reductions in prepulse inhibition of the startle reflex (PPI), GAD67 expression and cingulate 1 cortex volume.
[45]	Hungary	2013	Wister rat	Both male and female	Ketamine, 30 mg/kg 5 times/week, 15 injections in total) from PND35-PND56 of age, i.p. injection	After weaning at 3 weeks of age (PND21–23 days), Rats were housed individually for 28 days (between 4 and 7 weeks of age)	NaNo, NaTr, SelNo, and SelTr	Selective breeding after juvenile isolation and ketamine treatment produces several signs which resemble those found in schizophrenia.
[39]	Canada	2010	Sprague Dawley rat	Male	MK801, 0.5 mg/kg, (PND56-PND62) 2 × day for seven days, i.p. injection.	Social isolation started at postnatal day PND21 until PND56	GH + Sal, GH + MK, SI + Sal, and SI and MK	The lack of additive or synergistic effects in the “double hit model” suggests that combining isolation and subchronic MK801 treatment does not necessarily produce greater behavioural or physiological dysfunction than that seen with either treatment alone.
[50]	France	2020	Wister rat	Male	PCP (10 mg/kg) on (PND7, PND9, and PND11), s.c. injection	At the weaning day (PND 21), male rat pups were housed individually until the end of the study	V + GH + V, PCP + SI + V, and PCP + SI + Clo	The PCP-SI model presents with enduring and robust deficits (hyperactivity and social recognition impairment) associated with positive symptoms and cognitive/social deficits of schizophrenia, respectively. These deficits are normalized by chronic treatment with clozapine, thereby confirming the predictive validity of this animal model.
[38]	Hungary	2009	Wister rat	Male	Ketamine, 30 mg/kg, (PND28-PND42) one injection per day, i.p. injection	Wistar rats after weaning (PND21–PND23 days old) were either housed individually or grouped for 21 days.	GH + Sal, GH + Ket, SI + Sal, and SI + Ket	Since both social isolation and NMDA treatment are well-known animal models of schizophrenia, our results showed that juvenile isolation but not ketamine administration can stimulate hypoalgesia associated with this disease.
[34]	China	2017	Sprague Dawley rat	Male	MK801, 0.1, 0.3, and 0.5 mg/kg in PND7-PND21, s.c. injection	At PND21, rats were social isolated for four weeks (on PND49)	GH + Sal, SI + Sal, GH + MK0.1, SI + MK0.1, GH + MK0.3, SI + MK0.3, GH + MK0.5, and SI + MK0.5	Administration of MK801 and social isolation are two independent factors on the neurodevelopmental defects. Combining social isolation and subchronic MK801 treatment does not necessarily produce greater behavioural or physiological dysfunction than that seen with either treatment alone.
[40]	Canada	2010	Sprague Dawley rat	Male	MK801, 0.5 mg/kg, injected twice per day for 7 days from PND56-PND62, i.p. injection.	Rats were obtained at weaning (PND21); they were socially isolated, or group housed according to their randomly assigned housing groups and remained in their assigned groups for the duration of the experiment.	GH + Sal, GH + MK, SI + Sal, and SI + MK	Locomotor activity was increased in social isolated rats. This activity was exacerbated in MK801-SI rats suggesting a possible decrease in hippocampal and/or prefrontal cortex GABA function.
[46]	United Kingdom	2014	Lister hooded rat	Male	PCP, 10 mg/kg, on post-natal day (PND7, PND9, and PND11), s.c. injection	Rats were socially isolated on PND23, and animals remain isolated for 6 weeks.	GH + CTRL, GH + PCP, SI + CTRL, and SI + PCP	Neonatal PCP and social isolation both produced behavioural deficits in adult rats resulting in severe cognitive impairment (visual recognition memory impairment). This provided a comprehensive preclinical model that can be used to determine the neurobiological aetiology of schizophrenia than either treatment alone.
[43]	Canada	2013	Sprague Dawley rat	Male	MK801 0.5 mg/kg, (PND62-PND68) twice daily for 7 days, i.p. injection	Rats reared in groups or in isolation beginning at PND21.	GH + Sal, GH + MK, SI + Sal, and SI + MK	Results showed that polydipsia is a schizophrenia-like behavioural effect caused by social isolation. The “double hit” model did not yield a more pronounced polydipsia effect than each treatment alone.
[49]	United Kingdom	2020	Lister hooded rat	Male	PCP-HCL, 10 mg/kg on (PND 7, PND9, and PND11), s.c. injection	Animals were socially isolated on PND21-PND63.	GH + V, GH + PCP, SI + V, and SI + PCP	Glutamate release was reduced in a “double hit” model; this reduced interneuron firing and caused impairment in the novel object discrimination task.
[48]	China	2016	Sprague Dawley rat	Male	MK801, 0.2 mg/kg, (PND7-PND10), i.p. injected	Animals were socially isolated on PND21 and remained isolated for 8 weeks.	GH + Sal, GH + MK, and SI + Sal	Both socially reared rats with neonatal exposure to the NMDA receptor antagonist MK-801 and isolation-reared rats exhibited augmented startle responses.
[33]	Spain	2017	Transgenic strain mice	Male	MK801, 1 mg/kg on PND7, once off/one injection, i.p. injection	Rats were socially isolated on PND21 and remained isolated for 10 weeks.	GH + Sal, GH + MK, SI + Sal, and SI + MK	The “double hit” model showed that the change in E/I balance in the key brain regions as one of the underlying causes of schizophrenia.
[51]	Spain	2021	FVB mice	Male	MK801, 1 mg/kg on PND7, once off/one injection, i.p. injection	Rats were socially isolated on PND21 and remained isolated for 10 weeks.	GH + Sal, GH + MK, SI + Sal, and SI + MK	The “double hit” model showed a significant decrease in the number of PV+ interneurons, perineuronal nets (PNNs), and PNNs+PV+ cells when compared to control grouped mice.

Abbreviations: SI (social isolation), GH (group housed), MK (MK801), Sal (saline), Plz (phenelzine), L (lamotrigine), PCP (phencyclidine), CTRL (control), V (vehicle), THC (9-tetrahydrocannabinol), NaNo (naïve rats without treatment), NaTr (naïve rats with post-weaning social isolation and ketamine treatment), SelNo (selectively bred animals without any treatment), SelTr (selectively bred rats with both post-weaning social isolation and ketamine treatment), Clo (clozapine), Ket (ketamine), i.p. (intraperitoneal), s.c. (subcutaneous), PND (postnatal day).

## Data Availability

Included studies in the systematic review were obtained from [PubMed and Ebscohost].

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
