# Peer review of "Investigating the Robustness of a Rodent “Double Hit” (Post-Weaning Social Isolation and NMDA Receptor Antagonist) Model as an Animal Model for Schizophrenia: A Systematic Review"

_brainsci, 2023, doi:10.3390/brainsci13060848_

Round 1
Reviewer 1 Report
This is a very interesting paper reviewing the rodel "double hit" model as an animal model for schizophrenia. The paper is well written and of interest for the readers; however, several minor changes are recommended before further considering.
ABSTRACT
1- In the abstract section, I recommend to indicate if the authors followed the PRISMA statement. In the methods subsection, they describe that they used the MEDLINE, Google Scholar and Ebscohost databases.
2- I recommend to summarize the conclusions from the abstract.
INTRODUCTION
1- The authors start the introduction with a brief description of the clinical sumptoms of schizophrenia. I also recommend to add a brief paragraph about the treatment and prognosis, derived from these clinical symptoms.
2- The main aims are presented in the last sentence of the introduction. I recommend to expand the aims and objectives and to include them in a separate subsection (1.1- Aims).
METHODS
1- Is the systematic review following the PRISMA statement? Which guidelines did the authors use?
2-Why did the authors not include "basic research" within the search term strategy?
Flowchart: the records removed before screening should be indicated before records screened by title. Please, use the flowchart according to the PRISMA statement.
DISCUSSION
1- In the discussion section, the double hit model on positive symptoms and cognitive symptoms should be explained after the model on neurotransmitters.
2- I recommend to add a limitations and strenghts section.
Reviewer 2 Report
The paper reports a systematic review of the literature about the schizophrenia rodent model. The study included 16 papers and showed the presence of robust literature about the double hit theory for schizophrenia, while negative symptoms are quite neglected. The methods applied were correctly reported and they followed international guidelines. However, I think the authors should take into consideration some aspects to improve their paper:
- your literature search is quite old. Is there any reason? It was conducted more than two years ago and I think this is a big bias for your paper. Indeed, you might miss some papers reducing the value of your paper.
- Google Scholar is usually not used for systematic review because it has a lot of useless results. Is there any specific reason that you decided to use this database?
- Is there any comments about the animal model used?
- Have you any suggestions for the future?
- In table 3, is it possible to include the year of publication? It could be interesting to have this information easier.
- Please include a section with the limits of your manuscript.
- finally, all the tables need captions.
Round 2
Reviewer 2 Report
I think the authors did not take the opportunity to improve the manuscript but just justified what they had done, saying that they had problems and that they would cover the reported time gap with a new paper that would be very similar to the current one. In this perspective, the peer review process is useless. I think the authors should revise their manuscript, including the papers that might have been missed in these two years, and then evaluate a meta-analysis.
There are minor parts that might be improved.
